# AN INVESTIGATION OF MODEL-FREE PLANNING

## ABSTRACT

The field of reinforcement learning (RL) is facing increasingly challenging domains with combinatorial complexity. For an RL agent to address these challenges, it is essential that it can plan effectively. Prior work has typically utilized an explicit model of the environment, combined with a specific planning algorithm (such as tree search). More recently, a new family of methods have been proposed that learn how to plan, by providing the structure for planning via an inductive bias in the function approximator (such as a tree structured neural network), trained end-to-end by a model-free RL algorithm. In this paper, we go even further, and demonstrate empirically that an entirely model-free approach, without special structure beyond standard neural network components such as convolutional networks and LSTMs, can learn to exhibit many of the hallmarks that we would typically associate with a model-based planner. We measure our agent's effectiveness at planning in terms of its ability to generalize across a combinatorial and irreversible state space, its data efficiency, and its ability to utilize additional thinking time. We find that our agent has the characteristics that one might expect to find in a planning algorithm. Furthermore, it exceeds the state-of-the-art in challenging combinatorial domains such as Sokoban and outperforms other model-free approaches that utilize strong inductive biases toward planning.

## 1 INTRODUCTION

One of the aspirations of artificial intelligence is a cognitive agent that can adaptively and dynamically form plans to achieve its goal. Traditionally, this role has been filled by model-based RL approaches, which first learn an explicit model of the environment's system dynamics or rules, and then apply a planning algorithm (such as tree search) to the learned model. But model-based approaches have been challenging to scale with learned models in complex environments.

More recently, a variety of approaches have been proposed that learn to plan *implicitly*, solely by model-free training. These *model-free planning* agents utilize a special neural architecture that mirrors the structure of a particular planning algorithm. For example the neural network may be designed to represent search trees (Farquhar et al., 2017; Oh et al., 2017; Guez et al., 2018), forward simulations (Racanière et al., 2017; Silver et al., 2016), or dynamic programming (Tamar et al., 2016). The main idea is that, given the appropriate inductive bias for planning, the function approximator can learn to leverage these structures to learn its own planning algorithm. This kind of *algorithmic function approximation* may be more flexible than an explicit model-based approach, allowing the agent to customize the nature of planning to the specific environment.

In this paper we explore the hypothesis that planning may occur implicitly, even when the function approximator has no special inductive bias toward planning. Previous work (Pang & Werbos, 1998; Wang et al., 2018) have supported the idea that model-based behavior could be learned with general recurrent architectures, with planning computation amortized over multiple discrete steps (Schmidhuber, 1990), but comprehensive demonstrations of its effectiveness are still missing. Inspired by the successes of deep learning and the universality of neural representations, our main idea is simply to furnish a neural network with a high capacity and flexible representation, rather than mirror any particular planning structure. Given such flexibility, the network can in principle learn its own algorithm for approximate planning. Specifically, we utilize a family of neural networks based on a widely used function approximation architecture: the stacked convolutional LSTMs (ConvLSTM by Xingjian et al. (2015)).

It is perhaps surprising that a purely model-free reinforcement learning approach can be so successful in domains that would appear to necessitate explicit planning. This raises a natural question: what is planning? Can a model-free RL agent truly be considered to be planning, without any explicit model of the environment, and without any explicit simulation of that model? In this paper we take a behaviourist approach. Here, planning will rather be considered to be a measurable property of the agent's interactions. In particular, we consider three key properties that an agent equipped with planning should exhibit.

First, an effective planning algorithm should be able to generalize to different situations. The intuition here is that a simple function approximator cannot predict accurately across a combinatorial space of possibilities (for example the value of all chess positions), but a planning algorithm can perform a local search to dynamically compute predictions (for example by tree search). We measure this property using procedural environments (such as random gridworlds, Sokoban (Racanière et al., 2017), Boxworld (Zambaldi et al., 2018)) with a massively combinatorial space of possible layouts. We find that our model-free planning agent achieves state-of-the-art performance, and significantly outperforms more specialized model-free planning architectures. We also investigate extrapolation to a harder class of problems beyond those in the training set, and again find that our architecture performs effectively – especially with larger network sizes.

Second, a planning agent should be able to learn efficiently from relatively small amounts of data. Model-based RL is frequently motivated by the intuition that a model (for example the rules of chess) can often be learned more efficiently than direct predictions (for example the value of all chess positions). We measure this property by training our model-free planner on small data-sets, and find that our model-free planning agent still performs well and generalizes effectively to a held-out test set.

Third, an effective planning algorithm should be able to make good use of additional thinking time. Put simply, the more the algorithm thinks, the better its performance should be. This property is likely to be especially important in domains with irreversible consequences to wrong decisions (e.g. death or dead-ends). We measure this property in Sokoban by adding additional thinking time at the start of an episode, and find that our model-free planning agent solves considerably more problems.

Together, our results suggest that a model-free agent, without planning-inspired network structure, can learn to exhibit many of the behavioural characteristics of planning. The architecture presented in this paper serves to illustrate this point, and shows the surprising power of one simple approach. We hope our findings broaden the search for more general architectures that can tackle an even wider range of domains.

## 2 METHODS

We first motivate and describe the main network architecture we use in this paper. Then we briefly explain our training setup. More details can be found in Appendix C.

### 2.1 MODEL ARCHITECTURES

We desire models that can represent and learn powerful but unspecified planning procedures. Rather than encode strong inductive biases toward particular planning algorithms, we choose high-capacity neural network architectures that are capable of representing a very rich class of functions. As in many works in deep RL, we make use of convolutional neural networks (known to exploit the spatial structure inherent in visual domains) and LSTMs (known to be effective in sequential problems). Aside from these weak but common inductive biases, we keep our architecture as general and flexible as possible, and trust in standard model-free reinforcement learning algorithms to discover the functionality of planning.

#### 2.1.1 BASIC ARCHITECTURE

The basic element of the architecture is a ConvLSTM (Xingjian et al., 2015) – a neural network similar to an LSTM but with a 3D hidden state and convolutional operations. A recurrent network $f_\theta$ stacks together ConvLSTM modules. For a stack depth of $D$, the state $s$ contains all the cell states $c_d$ and outputs $h_d$ of each module $d$: $s = (c_1, \ldots, c_D, h_1, \ldots, h_D)$. The module weights

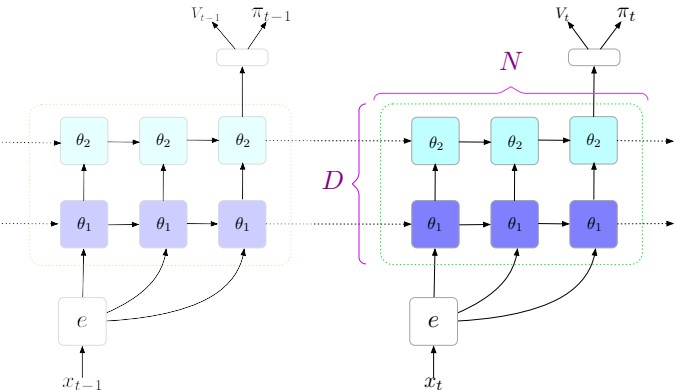

Figure 1: Illustration of the agent's network architecture. This diagram shows DRC(2,3) for two time steps. Square boxes denote ConvLSTM modules and the rectangle box represents an MLP. Boxes with the same color share parameters.

$\theta = (\theta_1, \ldots, \theta_D)$ are not shared along the stack. Given a previous state and an input tensor $i$, the next state is computed as $s' = f_\theta(s, i)$. The network $f_\theta$ is then repeated $N$ times within each time-step (i.e., multiple internal ticks per real time-step). If $s_{t-1}$ is the state at the end of the previous time-step, we obtain the new state given the input $i_t$ as:

$$s_t = g_\theta(s_{t-1}, i_t) = \underbrace{f_\theta(f_\theta(\ldots f_\theta(s_{t-1}, i_t), \ldots, i_t), i_t)}_{N \text{ times}} \tag{1}$$

The elements of $s_t$ all preserve the spatial dimensions of the input $i_t$. The final output $o_t$ of the recurrent network for a single time-step is $h_D$, the hidden state of the deepest ConvLSTM module after N ticks, obtained from $s_t$. We describe the ConvLSTM itself and alternative choices for memory modules in Appendix C.

The rest of the network is rather generic. An encoder network $e$ composed of convolutional layers processes the input observation $x_t$ into a $H \times W \times C$ tensor $i_t$ — given as input to the recurrent module $g$. The encoded input $i_t$ is also combined with $o_t$ through a skip-connection to produce the final network output. The network output is then flattend and an action distribution $\pi$ and a state-value $V$ are computed via a fully-connected MLP. The diagram in Fig 1 illustrates the full network.

From here on, we refer to this architecture as Deep Repeated ConvLSTM (DRC) network architecture, and sometimes followed explicitly by the value of $D$ and $N$ (e.g., DRC(3, 2) has depth $D = 3$ and $N = 2$ repeats).

### 2.1.2 ADDITIONAL DETAILS

Less essential design choices in the architectures are described here. Ablation studies show that these are not crucial, but do marginally improve performance (see Appendix F).

**Encoded observation skip-connection** The encoded observation $i_t$ is provided as an input to all ConvLSTM modules in the stack.

**Top-down skip connection** As described above, the flow of information in the network only goes up (and right through time). To allow for more general computation we add feedback connection from the last layer at one time step to the first layer of the next step.

**Pool-and-inject** To allow information to propagate faster in the spatial dimensions than the size of the convolutional kernel within the ConvLSTM stack, it is useful to provide a pooled version of the module's last output $h$ as an additional input on lateral connections. We use both max and mean pooling. Each pooling operation applies pooling spatially for each channel dimension, followed by a linear transform, and then tiles the result back into a 2D tensor. This is operation is related to the

pool-and-inject method introduced by Racanière et al. (2017) and to Squeeze-and-Excitation blocks (Hu et al., 2017).

**Padding** The convolutional operator is translation invariant. To help it understand where the edge of the input image is, we append a feature map to the input of the convolutional operators that has ones on the boundary and zeros inside.

## 2.2 TRAINING

We used a distributed framework to train an RL agent using the IMPALA V-trace actor-critic algorithm (Espeholt et al., 2018). While we found this training regime to help for training heavier networks, we also ran experiments which demonstrate that the DRC architecture can be trained effectively with A3C (Mnih et al., 2016). More details on the setup can be found in Appendix D.2.

## 3 PLANNING DOMAINS

Our domains are formally specified as RL problems, where agents must learn via reward feedback obtained by interacting with the environment (Sutton et al., 1998). We focus on combinatorial domains for which episodes are procedurally generated. In these domains each episode is instantiated in a pseudorandom configuration, so solving an episode typically requires some form of reasoning. Most of the environments are fully-observable and have simple 2D visual features. The domains are illustrated and explained in Appendix A. In addition to the planning domains listed below, we also run control experiments on a set of Atari 2600 games (Bellemare et al., 2013).

**Gridworld** A simple navigation domain following (Tamar et al., 2016), consisting of a grid filled with obstacles. The agent, goal, and obstacles are randomly placed for each episode.

**Sokoban** A difficult puzzle domain requiring an agent to push a set of boxes onto goal locations (Botea et al., 2003; Racanière et al., 2017). Irreversible wrong moves can make the puzzle unsolvable. We describe how we generate a large number of levels (for the fixed problem size 10x10 with 4 boxes) at multiple difficulty levels in Appendix B, and then each split into a training and test set. We are releasing these levels in the standard Sokoban format[1]. Unless otherwise specified, we ran experiments with the easier *unfiltered* set of levels.

**Boxworld** Introduced in (Zambaldi et al., 2018), the aim is to reach a goal target by collecting coloured keys and opening colour-matched boxes until a target is reached. The agent can see the keys (i.e., their colours) locked within boxes; thus, it must carefully plan the sequence of boxes that should be opened so that it can collect the keys that will lead to the target. Keys can only be used once, so opening an incorrect box can lead the agent down a dead-end path from which it cannot recover.

**MiniPacman** (Racanière et al., 2017). The player explores a maze that contains food while being chased by ghosts. The aim of the player is to collect all the rewarding food. There are also a few power pills which allow the player to attack ghosts (for a brief duration) and earn a large reward. See Appendix A.2 for more details.

## 4 RESULTS

We first examine the performance of our model and other approaches across domains. Then we report results aimed at understanding how elements of our architecture contribute to observed performance. Finally, we study evidence of iterative computation in Section 4.2 and generalization in Section 4.3.

## 4.1 COMPARISONS

In general, across all domains listed in Section 3, the DRC architecture performed very well with only modest tuning of hyper-parameters (see Appendix D). The DRC(3,3) variant was almost always the best in terms both of data efficiency (early learning) and asymptotic performance.

---

[1]Link to the dataset will be provided after the review process.

**Gridworld:** Many methods efficiently learn the Gridworld domain, especially for small grid sizes. We found that for larger grid sizes the DRC architecture learns more efficiently than a vanilla Convolutional Neural Network (CNN) architecture of similar weight and computational capacity. We also tested Value Iteration Networks (VIN) (Tamar et al., 2016), which are specially designed to deal with this kind of problem (i.e. local transitions in a fully-observable 2D state space). We found that VIN, which has many fewer parameters and a well-matched inductive bias, starts improving faster than other methods. It outperformed the CNN and even the DRC during early-stage training, but the DRC reached better final accuracy (see Table 1a and Figure 14a in the Appendix).

| Model | % solved at $1e6$ steps | % solved at $1e7$ steps |
|---|---|---|
| DRC(3, 3) | 30 | **99** |
| VIN | **80** | 97 |
| CNN | 3 | 90 |

(a)

| Model | % solved at $2e7$ steps | % solved at $1e9$ steps |
|---|---|---|
| DRC(3, 3) | **80** | **99** |
| ResNet | 14 | 96 |
| CNN | 25 | 92 |
| I2A (unroll=15) | 21 | 83 |
| 1D LSTM(3,3) | 5 | 74 |
| ATreeC | 1 | 57 |
| VIN | 12 | 56 |

(b)

Table 1: (a) Performance comparison in Gridworld, size 32x32, after 10M environment steps. VIN (Tamar et al., 2016) and experimental setup detailed in Appendix. (b) Comparison of test performance on (unfiltered) Sokoban levels for various methods. I2A (Racanière et al., 2017) results are re-rerun within our framework. ATreeC (Farquhar et al., 2017) results are detailed in Appendix. MCTSnets (Guez et al., 2018) also considered the same Sokoban domain but in an expert imitation setting (achieving 84% solved levels).

**Sokoban:** In Sokoban, we demonstrate state-of-the-art results versus prior work which targeted similar box-pushing puzzle domains (ATreeC (Farquhar et al., 2017), I2A (Racanière et al., 2017)) and other generic networks (LSTM (Hochreiter & Schmidhuber, 1997), ResNet (He et al., 2016), CNNs). We also test VIN on Sokoban, adapting the original approach to our state space by adding an input encoder to the model and an attention module at the output to deal with the imperfect state-action mappings. Table 1b compares the results for different architectures at the end of training. Only 1% of test levels remain unsolved by DRC(3,3) after 1e9 steps, with the second-best architecture (a large ResNet) failing four times as often.

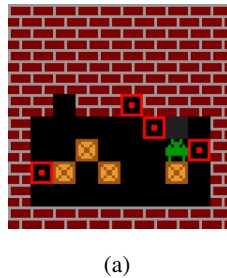 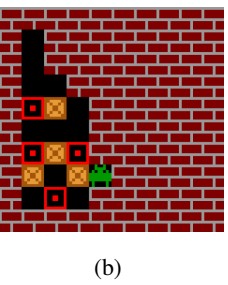 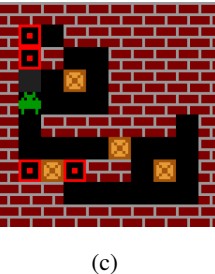

(a)        (b)        (c)

Figure 2: Examples of Sokoban levels from the (a) unfiltered, (b) medium test sets, and from the (c) hard set. Our best model is able to solve all three levels.

**Boxworld:** On this domain several methods obtain near-perfect final performance. Still, the DRC model learned faster than published methods, achieving ≈80% success after 2e8 steps. In comparison, the best ResNet achieved ≈50% by this point. The relational method of Zambaldi et al. (2018) can learn this task well but only solved <10% of levels after 2e8 steps.

**Atari 2600** To test the capacity of the DRC model to deal with richer sensory data, we also examined its performance on five planning-focussed Atari games (Bellemare et al., 2013). We obtained state-of-the-art scores on three of five games, and competitive scores on the other two (see Appendix E.2 and Figure 10 for details).

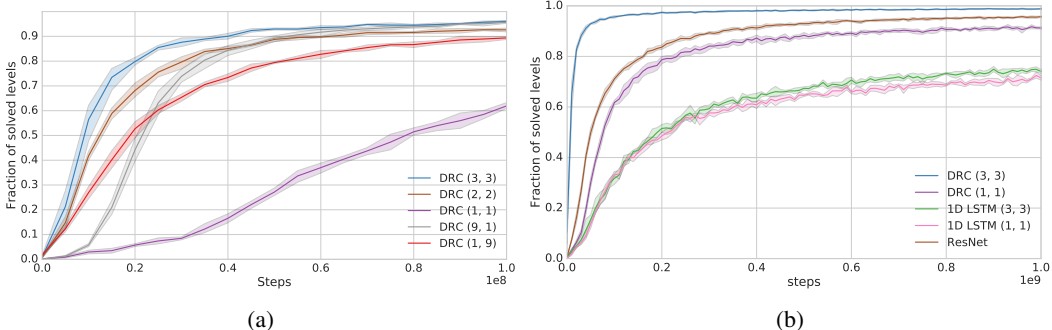

(a)                                          (b)

Figure 3: a) Learning curves for various configurations of DRC in Sokoban-Unfiltered. b) Comparison with other network architectures tuned for Sokoban. Results are on test-set levels.

### 4.1.1 INFLUENCE OF NETWORK ARCHITECTURE

We studied the influence of stacking and repeating the ConvLSTM modules in the DRC architecture, controlled by the parameters $D$ (stack length) and $N$ (number of repeats) as described in Section 2.1. These degrees of freedom allow our networks to compute its output using shared, iterative, computation with $N > 1$, as well as computations at different levels of representation and more capacity with $D > 1$. We found that the DRC(3,3) (i.e, $D = 3, N = 3$) worked robustly across all of the tested domain. We compared this to using the same number of modules stacked without repeats (DRC(9,1)) or only repeated without stacking (DRC(1,9)). In addition, we also look at the same smaller capacity versions $D = 2, N = 2$ and $D = 1, N = 1$ (which reduces to a standard ConvLSTM). Figure 3a shows the results on Sokoban for the different network configurations. In general, the versions with more capacity performed better. When trading-off stacking and repeating (with total of 9 modules), we observed that only repeating without stacking was not as effective (this has the same number of parameters as the DRC(1,1) version), and only stacking was slower to train in the early phase but obtained a similar final performance.

We also confirmed that DRC(3,3) performed better than DRC(1,1) in Boxworld, MiniPacman, and Gridworld (see Figure 15a in Appendix).

On harder Sokoban levels (Medium-difficulty dataset), we trained the DRC(3,3) and the larger capacity DRC(9,1) configurations and found that, even though DRC(9,1) was slower to learn at first, it ended up reaching a better score than DRC(3,3) (94% versus 91.5% after 1e9 steps). See Fig 9 in appendix. We tested the resulting DRC(9,1) agent on the hardest Sokoban setting (Hard-difficulty), and found that it solved 80% of levels in less than 48 minutes. In comparison, running a powerful tree search algorithm, Levin Tree Search (Orseau et al., 2018), with a DRC(1,1) as policy prior solves 94%, but in 10 hours.

In principle, deep feedforward models should support iterative procedures within a single time-step and perhaps match the performance of our recurrent networks (Jastrzebski et al., 2017). In practice, deep ResNets performed poorly versus our recurrent mod-

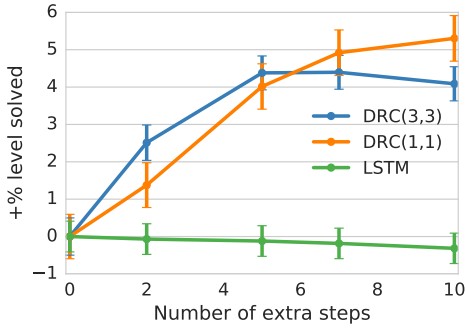

Figure 4: Forcing extra computation steps after training improves the performance of DRC on Sokoban-Medium set (5 networks, each tested on the same 5000 levels). Steps are performed by overriding the policy with no-op actions at the start of an episode.

els (see Figure 3b), and are in any case incapable of caching implicit iterative planning steps over time steps. Finally, we note that recurrence by itself was also not enough: replacing the ConvLSTM modules with flat 1-D LSTMs performed poorly (see Figure 3b).

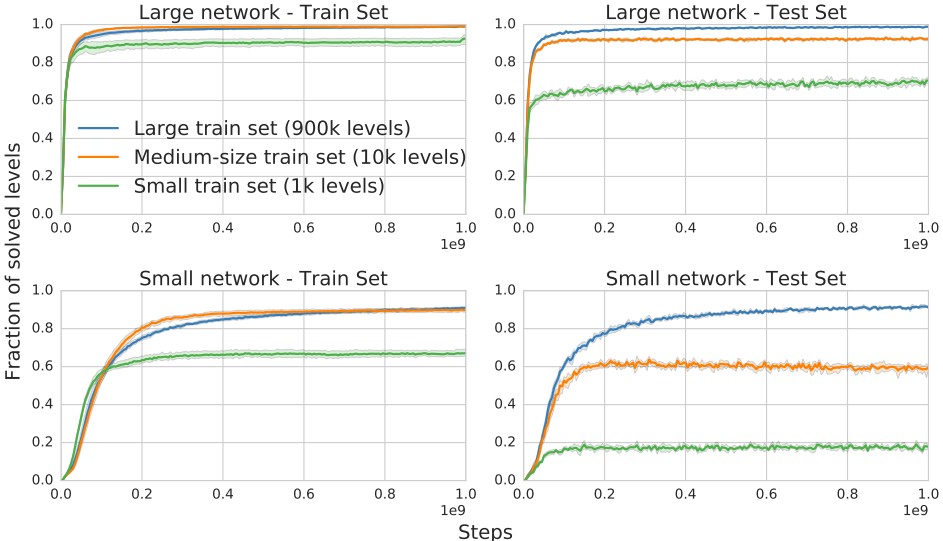

Figure 5: Comparison of DRC(3,3) (Top, Large network) and DRC(1,1) (Bottom, Small network) when trained with RL on various train set sizes (subsets of the Sokoban-unfiltered training set). Left column shows the performance on levels from the corresponding train set, right column shows the performance on the test set (the same set across these experiments).

Across experiments and domains, our results suggests that both the network capacity and the iterative aspect of a model drives the agent's performance.

## 4.2 ITERATIVE COMPUTATION

One desirable property for planning mechanisms is that their performance can scale with additional computation without seeing new data. Although RNNs (and more recently ResNets) can in principle learn a function that can be iterated to obtain a result (Graves, 2016; Jastrzebski et al., 2017; Greff et al., 2016), it is not clear whether the networks trained in our RL domains learn to amortize computation over time in this way. To test this, we took trained networks in Sokoban (unfiltered) and tested post hoc their ability to improve their results with additional steps. To do so we introduced 'no-op' actions at the start of each episode – up to 10 extra steps. We observed clear performance improvements on Medium difficulty levels of up to 5% for DRC networks (see Figure 4). We did not find such improvements for the simpler fully-connected LSTM architecture. This suggests that the networks have learned a scaleable strategy for the task which is computed and refined through a series of identical steps, thereby exhibiting one of the essential properties of a planning algorithm.

## 4.3 GENERALIZATION

In combinatorial domains generalization is a central issue. Given limited exposure to configurations in an environment, how well can a model perform on unseen scenarios? In the supervised setting, large flexible networks are capable of over-fitting. Thus, one concern when using high-capacity networks is that they may over-fit to the task, for example by memorizing, rather than learning a strategy that can generalize to novel situations. Recent empirical work in SL (Supervised Learning) has shown that the generalization of large networks is not well understood (Zhang et al., 2016; Arpit et al., 2017). Generalization in RL is even less well studied, though recent work by (Zhang et al., 2018a;b) has begun to explore the effect of training data diversity.

We explored two main axes in the space of generalization. We varied both the diversity of the environments as well as the size of our models. We trained the DRC architecture in various data regimes, by restricting the number of unique Sokoban levels — during the training, similar to SL, the training algorithm iterates on those limited levels many times. We either train on a Large (900k

levels), Medium (10k) or Small (1k) set. For each dataset size, we compared a larger version of the network, DRC(3,3), to a smaller version DRC(1,1) [2]. Results are shown in Figure 5.

In all cases, the larger DRC(3,3) network generalized better than its smaller counterpart, both in absolute terms and in terms of *generalization gap*. In particular, in the Medium regime, the generalization gap[3] is 6.482% for DRC(3,3) versus 33.398% for DRC(1, 1). Figure 6a shows the results when tested after training on both unfiltered and Medium test sets. We performed an analogous experiment in the Boxworld environment and observed remarkably similar results (see Appendix Fig 12).

Looking across these domains and experiments there are two findings that are of particular note. First, unlike analog SL experiments, reducing the number of training levels does not necessarily improve performance on the train set. Networks trained on 1k levels perform worse in terms of fraction of level solved. We believe this is due to the exploration problem in low-diversity regime: With more levels, the training agent faces a natural curriculum to help it progress toward harder levels. Another view of this is that larger networks can overfit the training levels, but only if they experience success on these levels at some point. While local minima for the loss in SL are not practically an issue, local minima in policy space can be problematic.

From a classic optimization perspective, a surprising finding is that the larger networks in our experiment (both Sokoban & Boxworld) suffer *less* from over-fitting in the low-data regime than their smaller counterparts (see Figure 6). However, this is in line with recent findings Zhang et al. (2016) in SL that the generalization of a model is driven by the architecture and nature of the data, rather than simply as a results of the network capacity and size of the dataset. Indeed, we also trained the same networks in a purely supervised fashion through imitation learning of an expert policy.[4] We observed a similar result when comparing the classification accuracy of the networks on the test set, with the DRC(3,3) better able to generalize — even though both networks had similar training errors on small datasets.

**Extrapolation**

Another facet of generality in the strategy found by the DRC network is how it performs outside the training distribution. In Sokoban, we tested the DRC(3,3) and DRC(1,1) networks on levels with a larger number of boxes than those seen in the training set. Figure 13a shows that DRC was able to extrapolate with little loss in performance to up to 7 boxes (for a a fixed grid size). The performance degradation for DRC(3,3) on 7 boxes was 3.5% and 18.5% for DRC(1,1). In comparison, the results from Racanière et al. (2017) report a loss of 34% when extrapolating to 7 boxes in the same setup.

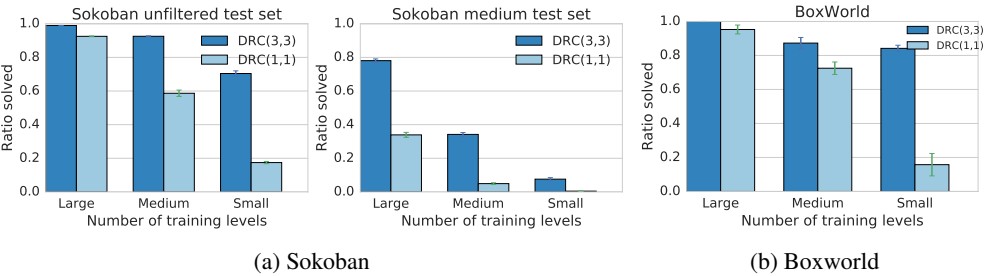

(a) Sokoban      (b) Boxworld

Figure 6: (a) Generalization results from a trained model on different training set size (Large, Medium and Small) for Sokoban. Left figure shows results on the unfiltered test set, right figure shows results on the medium test set. (b) Similar generalization results for trained models in Boxworld.

---

[2]DRC(3,3) has around 300K more parameters, and it requires around 3 times more computation

[3]We compute the generalization gap by subtracting the performance (ratio of levels solved) on the training set from performance on the test set.

[4]Data was sampled on-policy from the expert policy executed on levels from the training datasets.

## 5 DISCUSSION

We aspire to endow agents with the capacity to plan effectively in combinatorial domains where simple memorization of strategies is not feasible. An overarching question is regarding the nature of planning itself. Can the computations necessary for planning be learned solely using model-free RL, and this can be achieved by a general-purpose neural network with weak inductive biases? Or is it necessary to have dedicated planning machinery — either explicitly encoding existing planning algorithms, or implicitly mirroring their structure? In this paper, we studied a variety of different neural architectures trained using model-free RL in procedural planning tasks with combinatorial and irreversible state spaces. Our results suggest that general-purpose, high-capacity neural networks based on recurrent convolutional structure, are particularly efficient at learning to plan. This approach yielded state-of-the-art results on several domains – outperforming all of the specialized planning architectures that we tested. Our generalization and scaling analyses, together with the procedural nature of the studied domains, suggests that these networks learn an algorithm for approximate planning that is tailored to the domain. The algorithmic function approximator appears to compute its plan dynamically, amortised over many steps, and hence additional thinking time can boost its performance.

Recent work in the context of supervised learning is pushing us to rethink how large neural network models generalize (Zhang et al., 2016; Arpit et al., 2017). Our results further demonstrate the mismatch between traditional views on generalisation and model size. The surprising efficacy of our planning agent, when trained on a small number of scenarios across a combinatorial state space, suggests that any new theory must take into account the algorithmic function approximation capabilities of the model rather than simplistic measures of its complexity. Ultimately, we desire even more generality and scalability from our agents, and it remains to be seen whether model-free planning will be effective in reinforcement learning environments of real-world complexity.

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

## APPENDIX

The appendix is organized as follows: we first describe the five environments we used in this paper. We then provide details about the dataset generation process used for Sokoban dataset that we are releasing. Next, we describe the DRC architecture, various choices of memory type, and all the implementation details. We also list the parameters and experimental setup for our DRC all our models. We then present some extended experiments and extension of our generalization analysis, followed by ablation experiments for DRC and finally we compare the DRC against various baseline agents (VIN, ATreeC, and ResNet).

## A    DOMAINS

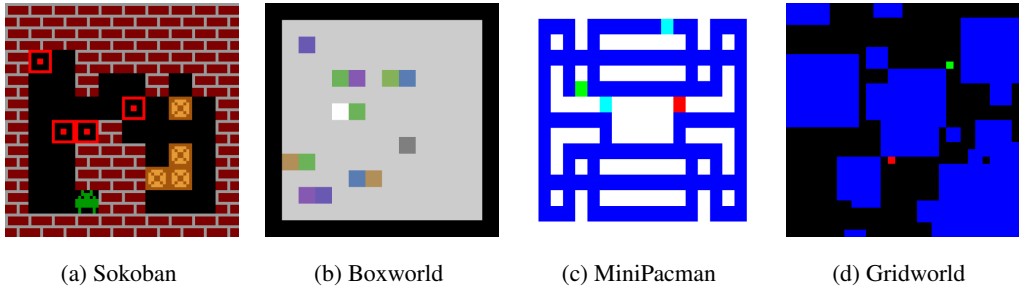

| (a) Sokoban | (b) Boxworld | (c) MiniPacman | (d) Gridworld |

Figure 7: Sample observations for each of the planning environments at the beginning of an episode.

### A.1    SOKOBAN

We follow the reward structure in Racanière et al. (2017), with a reward of 10 for completing a level, 1 for getting a box on a target, and -1 for removing a box from a target, in addition to a cost per time-step of -0.01. Each episode is capped at 120 frames. We use a sprite-based representation of the state as input. Each sprite is an (8, 8, 3)-shaped RGB image.

Dataset generation is described in Appendix B. Unless otherwise noted, we have used the levels from the unfiltered dataset in experiments.

### A.2    MINIPACMAN

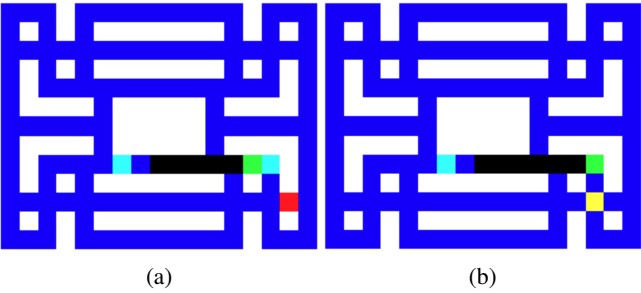

| (a) | (b) |

Figure 8: (a) Food in dark blue offers a reward of 1. After it is eaten, the food disappears, leaving a black space. Power pills in light blue offer a reward of 2. The player is in green, and ghosts in red. An episode ends when the player is eaten by a ghost. When the player has eaten all the food, the episode continues but the level is reset with power pills and ghosts placed at random positions. (b) When the player eats a power pill, ghosts turn yellow and become edible, with a larger reward or 5 for the player. They then progressively return to red (via orange), at which point they are once again dangerous for the player.

MiniPacman is a simplified version of the popular game Pac-Man. The first, deterministic version, was introduced in (Racanière et al., 2017). Here we use a stochastic version, where at every

time-step, each ghost moves with a probability of 0.95. This simulates, in a gridworld, the fact that ghosts move slower than the player. An implementation of this game is available under the name Pill Eater at `https://github.com/vasiloglou/mltrain-nips-2017/tree/master/sebastien_racaniere`, where 5 different reward structures (or *modes*) are available.

### A.3 BOXWORLD

In all our experiments we use branch length of 3 and maximum solution length of 4 . Each episode is capped at 120 frames. Unlike Sokoban, Boxworld levels are generated by the game engine. Unlike Sokoban we do not used sprite-based images; the environment state is provided as a (14, 14, 3) tensor.

### A.4 ATARI

We use the standard Atari setup of Mnih et al. (2015) with up to 30 no-ops at random, episodes capped at 30 mins, an action repeat of 4, and the full set of 18 actions. We made little effort to tune the hyperparameters for Atari; only the encoder size was increased. See Appendix E.2 for details about the experiments, and Figure 10 for the learning curves.

### A.5 GRIDWORLD

We generate 32x32-shaped levels by sampling a number of obstacles between 12 and 24. Each obstacle is a square, with side in [2, 10]. These obstacles may overlap. The player and goal are placed on random different empty squares. We use rejection sampling to ensure the player can reach the goal. Episodes terminate when reaching a goal or when stepping on an obstacle, with a reward of 1 and -1 respectively. There is a small negative reward of -0.01 at each step otherwise.

## B DATASET GENERATION DETAILS

The unfiltered Sokoban levels are generated by the procedure described in Racanière et al. (2017). Simple hashing is used to avoid repeats. To generate medium levels, we trained a DRC(1,1) policy whose LSTM state is reset at every time step. It achieves around 95% of level solved on the easy set. Then, we generate medium difficulty levels by rejection sampling: we sample levels using the procedure from Racanière et al. (2017), and accept a level if the policy cannot solve it after 10 attempts. The levels are not subsampled from the easy levels dataset but from fresh generation instead, so overlap between easy and medium is minimal (though there are 120 levels in common in the easy and medium datasets). To generate hard levels, we train a DRC(3,3) policy on a mixture of easy and medium levels, until it reached a performance level of around 85%. We then select the first 3332 levels of a separate medium dataset which are not solved by that policy after 10 attempts (so there is no overlap between hard and medium sets).

| Unfiltered | Medium | Hard |
|---|---|---|
| 99% | 95% | 80% |

Table 2: Best results obtained on the test set for the different difficulty levels across RL agents within 2e9 steps of training (averaged across 5 independent runs).

## C MODEL

We denoted $g_\theta(s_{t-1}, i_t) = f_\theta(f_\theta(\ldots f_\theta(s_{t-1}, i_t), \ldots, i_t), i_t)$ as the computation at a full time-step of the DRC(D, N) architecture. Here $\theta = (\theta_1, \ldots, \theta_D)$ are the parameters of $D$ stacked memory modules, and $i_t = e(x_t)$ is the agent's encoded observation.

Let $s_{t-1}^1, \ldots, s_{t-1}^N$ be the state of the $D$ stacked memory modules at the $N$ ticks at time-step $t - 1$. Each $s_{t-1}^n = (c_1^n, \ldots, c_d^n, h_1^n, \ldots, h_d^n)$. We then have the "repeat" recurrence below:

|  | Unfiltered train | Unfiltered test | Medium train | Medium test | hard |
|---|---|---|---|---|---|
| Unfiltered train | 900,000 | 0 | 119 | 10 | 0 |
| Unfiltered test | 0 | 100,000 | 10 | 1 | 0 |
| Medium train | 119 | 1 | 450,000 | 0 | 0 |
| Medium test | 10 | 0 | 0 | 50,000 | 0 |
| Hard | 0 | 0 | 0 | 0 | 3332 |

Table 3: Number of levels in each subset of the dataset and number of overlaps between them.

$$
\begin{aligned}
s_{t-1}^1 &= f_\theta(s_{t-1}, i_t) \\
s_{t-1}^n &= f_\theta(s_{t-1}^{n-1}, i_t) \text{ for } 1 < n \le N \\
s_t &= s_{t-1}^N
\end{aligned}
\tag{2}
$$

We can now describe the computation within a single tick, i.e., outputs $c_d^n$ and $h_d^n$ at $d = 1, \ldots, D$ for a fixed $n$. This is the "stack" recurrence:

$$
c_d^n, h_d^n = \text{MemoryModule}_{\theta_d}(i_t, c_d^{n-1}, h_d^{n-1}, h_{d-1}^n)
\tag{3}
$$

Note that the encoded observation $i_t$ is fed as an input not only at all ticks $1, \ldots, N$, but also at all depths $1, \ldots, D$ of the memory stack. (The latter is described in Section 2.1.2 but not depicted in Figure 1 for clarity.)

Generally each memory module is a ConvLSTM parameterized by $\theta_d$ at location (d, n) of the DRC grid. But we describe alternative choices of memory module in Appendix C.2.

## C.1 ADDITIONAL DETAILS/IMPROVEMENTS

### C.1.1 TOP-DOWN SKIP CONNECTION

As described in the stack recurrence, we feed the hidden state of each memory module as an input $h_{d-1}^n$ to the next module in the stack. But this input is only available for modules at depth $d > 1$. Instead of setting $h_0^n = \mathbf{0}$, we use a top-down skip connection i.e. $h_0^n = h_D^{n-1}$. We will explore the role of this skip-connection in Appendix F.

### C.1.2 POOL-AND-INJECT

The pool operation aggregates hidden states across their spatial dimensions to give vectors $m_1^n, \ldots, m_D^n$. We project these through a linear layer each (weights $W_{p_1}, \ldots, W_{p_D}$), and then tile them over space to obtain summary tensors $p_1^n, \ldots, p_D^n$. These have the same shapes as the original hidden states, and can be injected as additional inputs to the memory modules at the next tick.

$$
\begin{aligned}
m_d^n &:= [\max_{H,W}(h_d^n), \text{mean}_{H,W}(h_d^n)]^T \\
p_d^n &:= \text{Tile}_{H,W}(W_{p_d} m_d^n)
\end{aligned}
\tag{4}
$$

Finally, $p_d^{n-1}$ is provided as an additional input at location (d,n) of the DRC, and equation C becomes:

$$
c_d^n, h_d^n = \text{MemoryModule}_{\theta_d}(i_t, c_d^{n-1}, h_d^{n-1}, h_{d-1}^n, p_d^{n-1})
$$

## C.2 MEMORY MODULES

### C.2.1 CONVLSTM

$$c_d^n, h_d^n = \text{ConvLSTM}_{\theta_d}(i_t, c_d^{n-1}, h_d^{n-1}, h_{d-1}^n)$$

For all $d > 1$:

$$f_d^n = \sigma(W_{fi} * i_t + W_{fh_1} * h_{d-1}^n + W_{fh_2} * h_d^{n-1} + b_f)$$

$$i_d^n = \sigma(W_{ii} * i_t + W_{ih_1} * h_{d-1}^n + W_{ih_2} * h_d^{n-1} + b_i)$$

$$o_d^n = \sigma(W_{oi} * i_t + W_{oh_1} * h_{d-1}^n + W_{oh_2} * h_d^{n-1} + b_o)$$

$$c_d^n = f_d^n \odot c_d^{n-1} + i_d^n \odot \tanh(W_{ci} * i_t + W_{ch_1} * h_{d-1}^n + W_{ch_2} * h_d^{n-1} + b_c)$$

$$h_d^n = o_d^n \odot \tanh(c_d^n)$$

For $d = 1$, we use the top-down skip connection $h_D^{n-1}$ in place of $h_{d-1}^n$ as described above.

Here $*$ denotes the convolution operator, and $\odot$ denotes point-wise multiplication. Note that $\theta_d = (W_{f.}, W_{i.}, W_{o.}, W_{c.}, b_f, b_i, b_o, b_c)_d$ parameterizes the computation of the forget gate, input gate, output gate, and new cell state. $i_d^n$ and $o_d^n$ should not be confused with the encoded input $i_t$ or the final output $o_t$ of the entire network.

### C.2.2 GATEDCONVRNN

This is a simpler module with no cell state.

$$h_d^n = \text{GatedConvRNN}_{\theta_d}(i_t, h_d^{n-1}, h_{d-1}^n)$$

For all $d > 1$:

$$o_d^n = \sigma(W_{oi} * i_t + W_{oh_1} * h_{d-1}^n + W_{oh_2} * h_d^{n-1} + b_o)$$

$$h_d^n = o_d^n \odot \tanh(W_{hi} * i_t + W_{hh_1} * h_{d-1}^n + W_{hh_2} * h_d^{n-1} + b_h)$$

$\theta_d = (W_{o.}, W_{h.}, b_o, b_h)_d$ has half as many parameters as in the case of the ConvLSTM.

### C.2.3 SIMPLECONVRNN

This is a classic RNN without gating.

$$h_d^n = \tanh(W_{hi} * i_t + W_{hh_1} * h_{d-1}^n + W_{hh_2} * h_d^{n-1} + b_h)$$

$\theta_d = (W_{h.}, b_h)_d$ has half as many parameters as the GatedConvRNN, and only a quarter as the ConvLSTM.

We will explore the difference between these memory types in Appendix F. Otherwise, we use the ConvLSTM for all experiments.

## D HYPER-PARAMETERS

We tuned our hyper-parameters for the DRC models on Sokoban and used the same settings for all other domains. We only changed the encoder network architectures to accommodate the specific observation format of each domain.

### D.1 NETWORK PARAMETERS

All activation functions are ReLU unless otherwise specified.

**Observation size**

- Sokoban: (80, 80, 3)
- Boxworld: (14, 14, 3)
- MiniPacman: (15, 19, 3)
- Atari: (210, 160, 3)

**Encoder network** is a multilayer CNN with following parameters for each domain:

- Sokoban: number of channels: (32, 32), kernel size: (8, 4), strides: (4, 2)
- BoxWorld: number of channels: (32, 32), kernel size: (3, 2), strides: (1, 1)
- Mini-Pacman: number of channels: (32, 32), kernel size: (3, 3), strides: (1, 1)
- Gridworld: number of channels: (64, 64, 32), kernel size: (3, 3, 2), strides: (1, 1, 2)
- Atari: number of channels: (16, 32, 64, 128, 64, 32), kernel size: (8, 4, 4, 3, 3, 3), strides: (4, 2, 2, 1, 1, 1)

**DRC(D, N)**: all ConvLSTM modules use one convolution with 128 output channels, a kernel size of 3 and stride of 1. All cell states and hidden states have 32 channels.

**1D LSTM(D, N)**: all LSTM modules have hidden size 200.

**Policy**: single hidden layer MLP with 256 units.

**Total number of parameters** for all our models are shown in Figure 4.

| Model | Number of parameters |
|---|---|
| DRC(3,3) | 2,023,174 |
| DRC(1,1) | 1,746,054 |
| 1D LSTM (3,3) | 2,151,790 |
| CNN | 2,089,222 |
| ResNet for Sokoban | 52,584,982 |
| ResNet for Boxworld | 3,814,197 |
| ResNet for MiniPacman | 2,490,902 |
| I2A | 7,781,269 |
| VIN | 61,224 |
| ATreeC | 3,269,063 |

Table 4: Number of parameters for various models

### D.2 RL TRAINING SETUP

We use the V-trace actor-critic algorithm described by Espeholt et al. (2018), with 4 GPUs for each learner and 200 actors generating trajectories. We reduce variance and improve stability by using $\lambda$-returns targets ($\lambda = 0.97$) and a smaller discount factor ($\gamma = 0.97$). This marginally reduces the maximum performance observed, but increases the stability and average performance across runs, allowing better comparisons. For all experiments, we use a BPTT (Backpropagation Through Time) unroll of length 20 and a batch size of 32. We use the Adam optimizer (Kingma & Ba, 2014). The learning rate is initialized to 4e-4 and is annealed to 0 over 1.5e9 environment steps with polynomial annealing. The other Adam optimizer parameters are $\beta_1 = 0.9, \beta_2 = 0.999, \epsilon =$1e-4. The entropy and baseline loss weights are set to 0.01 and 0.5 respectively. We also apply a $\mathcal{L}^2$ norm cost with a weight of 1e-3 on the logits, and a $\mathcal{L}^2$ regularization cost with a weight of 1e-5 to the linear layers that compute the baseline value and logits.

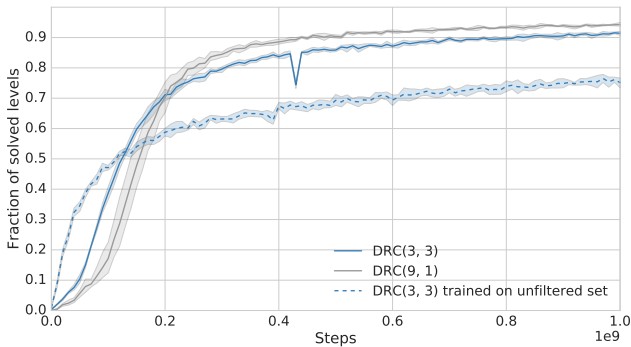

Figure 9: Learning curves in Sokoban for DRC architectures tested on the medium-difficulty test set. The dashed curve shows DRC(3,3) trained on the easier unfiltered dataset. The solid curves show DRC(3,3) and DRC(9,1) trained directly on the medium-difficulty train set.

# E    EXTRA EXPERIMENTS

## E.1    SOKOBAN

Figure 9 shows learning curve when different DRC architectures have been trained on the medium set as opposed to unfiltered set in the main text.

## E.2    ATARI

We train DRC(3,3) and DRC(1,1) on five Atari 2600 games: *Alien*, *Asteroid*, *Breakout*, *Ms. Pac-Man*, and *Up'n Down*. We picked these for their planning focus. Figure 10 shows learning curves comparing DRC(3,3) against DRC(1,1) and ApeX-DQN (Horgan et al., 2018) on the above games. We improve on ApeX-DQN on three out of five games.

Our scores generally improve with more training. For example, if we let the DRC(3,3) run for longer, it reaches scores above 75000 in Ms. Pacman.

# F    EXTRA ABLATION STUDIES

In Figure 11, we compare the Sokoban test-set performance of our baseline DRC(3,3) agent against various ablated versions. These results justify our choice of the ConvLSTM memory module and the architectural improvements described in Section 2.1.2.

## F.1    MEMORY TYPE

The ConvLSTM is the top-performing memory module among those we tested. The Gated Con-vRNN module comes out very close with half as many parameters. We surmise that the gap between these two types could be larger for other domains.

The Simple ConvRNN suffers from instability and high variance as expected, asserting that some gating is essential for the DRC's performance.

## F.2    ADDITIONAL IMPROVEMENTS

Without pool-and-inject, the network learns significantly slower at early stages but nevertheless converges to the same performance as the baseline. The vision shortcut seems to have little influence on the baseline model. This is likely because we already feed the encoded observation to every (d,n)-location of the DRC grid (see Appendix C). Without the top-down skip connection, the model exhibits larger performance variance and also slightly lower final performance. On the whole, none of these are critical to the model's performance.

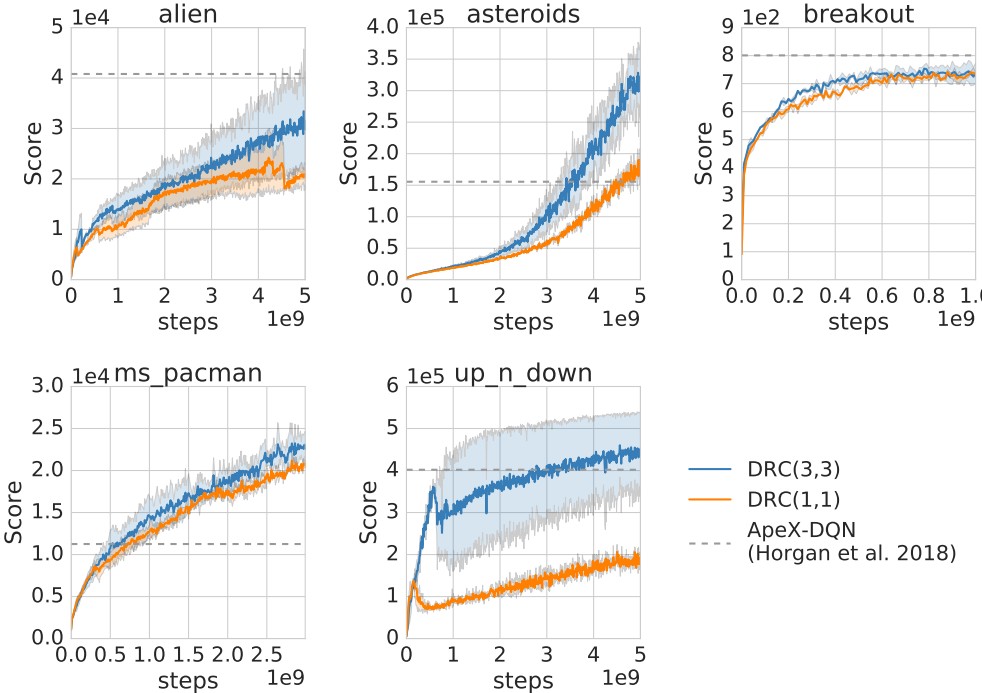

Figure 10: Learning curves comparing the DRC(3,3) and DRC(1,1) network configurations in 5 Atari 2600 games. Results are averaged over two independent runs. We also provide ApeX-DQN (no-op regime) results from Horgan et al. (2018) as a reference.

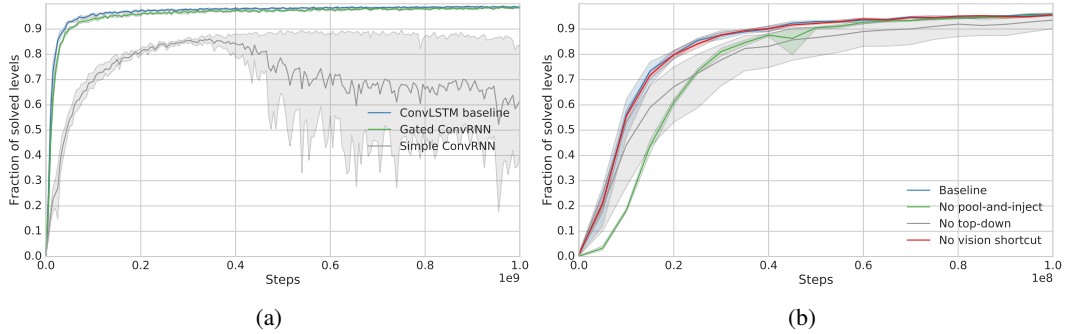

Figure 11: Ablation studies on the performance of our baseline DRC(3,3) agent on Sokoban. All curves show test-set performance. a) We replace the ConvLSTM for simpler memory modules. b) We remove our extra implementation details, namely pool-and-inject, the vision shortcut, and the top-down skip connection, from the model.

## G GENERALIZATION

For generalization experiments in Boxworld we generate levels online for each level-set size using seeds. Large approximately corresponds to 22k levels, medium to 5k, and small to 800. But at each iteration it's guaranteed that same levels are generated. Figure 12 shows results of similar experiment as Figure 5 but for BoxWorld domain.

Figure 13a shows the extrapolation results on Sokoban when a model that is trained with 4 boxes is tested on levels with larger number of boxes. Performance degradation for DRC(3,3) is very minimal and the models is still able to perform with performance of above 90% even on the levels with 7 boxes, whereas the performance on for DRC(1,1) drops to under 80%.

Figure 13b shows the generalization gap for Sokoban. Generalization gap is computed by the difference of performance (ratio of the levels solved) between train set and test set. The gap increase substantially more for DRC(1,1) compared to DRC(3,3) as the training set size decreases.

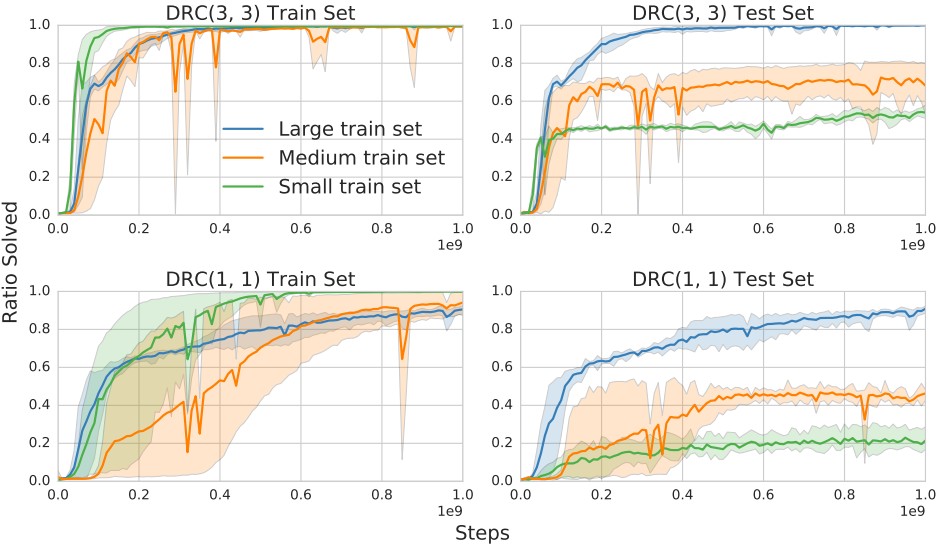

Figure 12: Generalization performance on BoxWorld when model is trained on different dataset sizes

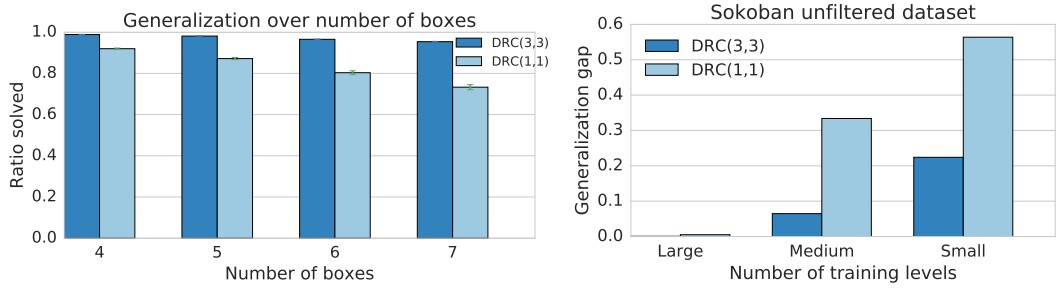

(a) Extrapolation over number of boxes.    (b) Generalization gap for different train set size.

Figure 13: (a) Extrapolation to larger number of boxes than those seen in training. The model was trained on 4 boxes. (b) Generalization gap computed as the difference between performance (in % of level solved) on train set against test set. Smaller bars correspond to better generalization.

## H  BASELINE ARCHITECTURES AND EXPERIMENTS

### H.1  VALUE ITERATION NETWORKS

Our implementation of VIN closely follows (Tamar et al., 2016). This includes using knowledge of the player position in the mechanism for attention over the q-values produced by the VIN module in the case of Gridworld.

For Sokoban, which provides pixel observations, we feed an encoded frame $I$ to the VIN module. The convolutional encoder is identical to that used by our DRC models (see Appendix D.1). We also couldn't use the player position directly; instead we use a learned soft attention mask $A = \sigma(W_{attention} * I)$ over the 2D state space, and sum over the attended values $A \odot \bar{Q}_a$ to obtain an attention-modulated value per abstract action $a$.

We introduce an additional choice of update rule in our experiments. The original algorithm sets $\bar{Q}_a{}' = W_{transition}^a * [\bar{R} : \bar{V}]$ at each value iteration. We also try the following alternative: $\bar{Q}_a{}' = \bar{R} + \gamma(W_{transition}^a * \bar{V})$ (note that ':' denotes concatenation along the feature dimension and '*' denotes 2D convolution). Here, $\gamma$ is a discounting parameter independent of the MDP's discount factor. In both cases, $\bar{R} = W_{reward} * I$ and $\bar{V} = \max_a \bar{Q}_a$. Although these equations denote a single convolutional operation in the computation of $\bar{Q}_a$, $\bar{R}$, and the attention map A, in practice we use two convolutional layers in each case with a relu activation in between.

We performed parameter sweeps over the choice of update rules (including $\gamma$), learning rate initial value, value iteration depth, number of abstract actions, and intermediate reward channels. The annealing schedule for the learning rate was the same as described in Appendix D.2. In total we had 64 parameter combinations for Gridworld, and 102 for Sokoban. We show the average of the top five performing runs from each sweep in Figures 14a and 14b.

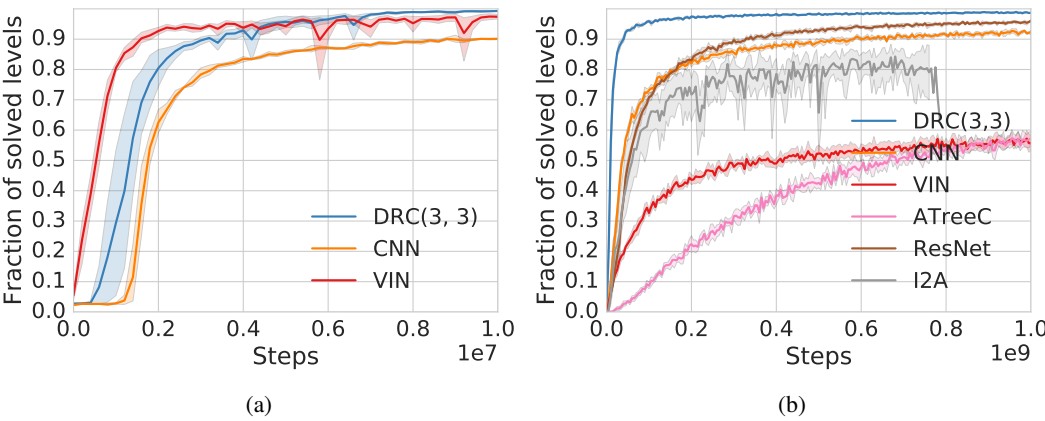

(a)                                                                 (b)

Figure 14: Performance curves comparing DRC(3,3) against baseline architectures on (a) Gridworld (32x32) and (b) Sokoban. The Sokoban curves show the test-set performance. For VIN we average the top five performing curves from the parameter sweep on each domain. For the other architectures we follow the approach described in Appendix I, averaging not the top five curves but five independent replicas for fixed hyperparameters.

### H.2  ATREEC

We implement and use the actor-critic formulation of (Farquhar et al., 2017). We feed it similarly encoded observations from Sokoban as in our DRC and VIN setups. This is flattened and passed through a fully connected layer before the tree planning and backup steps. We do not use any auxiliary losses for reward or state grounding.

We swept over the learning rate initial value, tree depth (2 or 3), embedding size (128, 512), choice of value aggregation (max or softmax), and TD-lambda (0.8, 0.9). We average the top five best-performing runs from the sweep in Figure 14b.

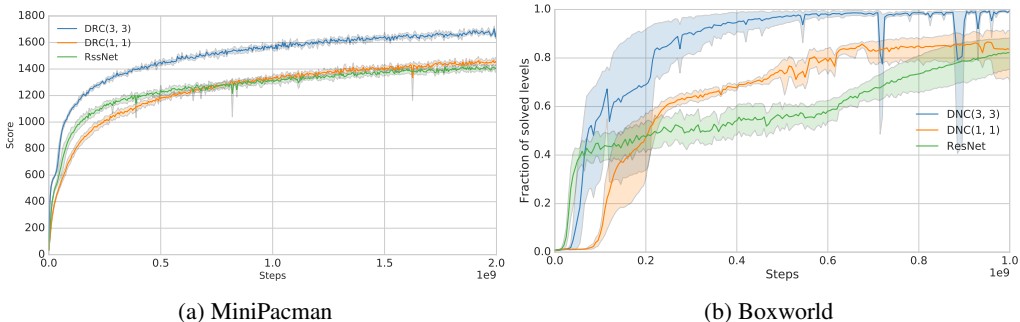

(a) MiniPacman                      (b) Boxworld

Figure 15: Performance curves comparing DRC(3,3) and DRC(1,1) against the best ResNet architectures we found for (a) MiniPacman and (b) Boxworld. DRC(3,3) reaches nearly 100% on Boxworld at $1e9$ steps.

## H.3 RESNET

We experiment with different ResNet architectures for each domain and choose one that performs best.

**Sokoban** Each layer was constructed by one CNN layer that could be potentially used for down-sampling using the stride parameter, and two residual blocks on top which consisted of one CNN layer. We used 9 of the described layers with (32, 32, 64, 64, 64, 64, 64, 64, 64) channels, (8, 4, 4, 4, 4, 4, 4, 4, 4) kernel shapes and (4, 2, 1, 1, 1, 1, 1, 1, 1) strides for the down-sampling CNN. And the output flattened and passed to 1 hidden layer MLP with 256 units, before computing the policy and baseline.

**Boxworld** We had a three layer CNN with (16, 32, 64) channels, kernel shape of 2 and stride of 1. And on top of that 8 ResNet blocks. Each blocks was consisted of the 2 layer CNN with 64 channels, kernel shape of 3 and stride 1. And the output flattened and passed to 1 hidden layer MLP with 256 units, before computing the policy and baseline.

**MiniPacman** We used the same model architecture as the one used for Boxworld with only difference being the initial CNN layer channels were (16, 32, 32) and ResNet blocks had 32 channels.

Figures 15a and 15b compare the ResNets above against DRC(3,3) and DRC(1,1) on MiniPacman and Boxworld.

## H.4 CNN

The CNN network we use is similar to the encoder network for DRC with more layers. It has (32, 32, 64, 64, 64, 64, 64, 64, 64) channels, (8, 4, 4, 4, 4, 4, 4, 4, 4) kernel sizes and strides of (4, 2, 1, 1, 1, 1, 1, 1, 1).

## H.5 I2A

We use the I2A implementation available at `https://github.com/vasiloglou/mltrain-nips-2017/tree/master/sebastien_racaniere` with the following modifications:

- We replace the FrameProcessing class with a 3-layer CNN with kernel sizes $(8, 3, 3)$, strides $(8, 1, 1)$ and channels $(32, 64, 64)$; the output of the last convolution is then flattened and passed through a linear layer with $512$ outputs.

- The model-free path passed to the I2A class was a FrameProcessing as describe above.

- The RolloutPolicy is a 2-layer CNN with kernel sizes $(8, 1)$, strides $(8, 1)$ and channels $(16, 16)$; the output of the last convolution is flattened and passed through an MLP with one hidden layer of size 128, and output size 5 (the number of actions in Sokoban).

- The environment model is similar to the one used in Racanière et al. (2017). It consists of a deterministic model that given a frame and one-hot encoded action, outputs a predicted next frame and reward. The input frame is first reduced in size using a convolutional layer with kernel size 8, stride 8 and 32 channels. The action is then combined with the output of this layer using pool-and-inject (see (Racanière et al., 2017)). This is followed by 2 size preserving residual blocks with convolutions of shape 3. The output of the last residual block is then passed to two separate networks to predict frame and reward. The frame prediction head uses a convolution layer with shape 3, stride 1, followed by a de-convolution with stride 8, kernel shape 8 and 3 channels. The reward prediction head uses a convolution layer with shape 3, stride 1, followed by a linear layer with output size 3. We predict rewards binned in 3 categories (less than $-1$, between $[-1, 1]$ and greater than 1), reducing it to a classification problem. Racanière et al. (2017).

The agent was trained with the V-trace actor-critic algorithm described by Espeholt et al. (2018). Unlike in Racanière et al. (2017), the environment model was not pre-trained. Instead it was trained online, using the same data that was used to train the RL loss.

## I  DETAILS ABOUT FIGURES

Unless otherwise noted, each curve is the average of five independent runs (with identical parameters). We also show a 95% confidence interval (using a shaded area) around each averaged curve. Before independent runs are averaged, we smoothen them with a window size of 5 million steps. Steps in RL learning curves refer to the total number of environment steps seen by the actors.

