# OpenReview forum: "An investigation of model-free planning"
_ICLR.cc/2019/Conference_

### Official Review · AnonReviewer2 · 2018-10-31
**Is it really planning, or something else?**

**Rating:** 4
**Confidence:** 5

**Review:**

The authors hypothesize that, under appropriate conditions, neural networks without specific architectural biases trained by model-free reinforcement learning algorithms are capable of learning procedures that are analogous to planning. This is certainly an important area of research in reinforcement learning.

Unfortunately, the approach employed to demonstrate this hypothesis seems flawed, which is why this submission should be rejected in its present form.

The authors suggest that the presence of planning should be accompanied by three observable characteristics: generalization of desired behavior to radically different situations, learning of desired behavior from small amounts of data, and ability to benefit from additional "thinking" time. Instead of trying to identify how an environmental model is represented by a network and how it is used for planning, the authors focus on checking for the aforementioned characteristics.

Even after conceding their strong claim despite weak argumentation provided by the authors, there are fundamental experimental issues that make the conclusions of this study unwarranted. Regarding the first two characteristics, the concepts of "radically different situations" or "small amounts of data" are extremely vague. Basically the authors assume that their problems are difficult enough to require planning. Having solved these problems with their proposed architecture, they conclude that planning must have occurred. Regarding the use of additional "thinking time," the authors claim that the improvement in performance caused by providing additional micro-steps to a recurrent neural network is clear evidence that something analogous to planning is happening, which is obviously not the case.

While it would not be surprising if there was indeed something analogous to planning happening inside the networks under consideration, this paper presents no stronger evidence for this claim than most other deep reinforcement learning papers that tackle complex environments.

Perhaps the most important contribution of this submission is the architecture based on ConvLSTMs proposed by the authors, which apparently surpasses many alternatives, including some biased towards planning. However, surpassing planning models is not strong evidence of planning. When stripped of unwarranted claims made by the authors regarding implicit planning, the proposed architecture does not seem sufficiently novel to warrant acceptance.

The authors should be commended for what was certainly very demanding experimental work, even though it does not support their core claims. Their second most important contribution is the experimental comparison between several recent architectures in a diverse selection of environments.

The writing is clear and accessible, except possibly for the architectural details described in Section 2.1.2, which do not seem very important. There are also several typos in Appendix D.2.

Regarding related work, the authors mention that Pang and Werbos [1] "advanced the approach." But they do not explain how they advanced this approach. In fact, we could not find much about this in the 1998 paper. Also, to our knowledge, "additional thinking time" was first proposed in the context of reinforcement learning and planning with two interacting RNNs by Schmidhuber [2, Section: "more network ticks than environmental ticks"].

[1] Xiaozhong Pang & Paul J. Werbos (1998): Neural Network Design for J Function Approximation in Dynamic Programming
[2] J.  Schmidhuber. Making the world differentiable: On using fully recurrent self-supervised neural networks for dynamic reinforcement learning and planning in non-stationary environments. TR FKI-126-90, TU Munich, November 1990.

Perhaps a strongly revised version the paper might become more acceptable if the authors addressed the issues above and especially toned down their claims about having experimentally identified the emergence of planning. Instead they should be extremely careful here, perhaps present this as "intriguing results," and address all possible counter arguments.

---

> ### Author Response · Authors · 2018-11-16
> **Response to AnonReviewer2**
>
> Thank you for your thoughtful and detailed review. We have written a common response to all three reviewers in a separate comment addressing the common issues. Below we address specific issues raised by AnonReviewer2.
>
> -“How environmental model is represented by a network”. We  emphasize that our agent is model-free, and hence it is not straightforward to assess if and how an environment model is represented in some form by the network. We can and do measure the network’s behavioural capacity to generalize to new scenarios, and in this sense we find it surprisingly flexible. It is unclear to what extent this indicates that the network has learned an implicit environment model.  This would be difficult to test, though it would be an interesting question for future work.
>
> Regarding the difficulty of tasks, the released Sokoban levels will be available after deanonymization of the submission. Some of the levels are quite challenging for human players. Anecdotally, there are many problems solved by the agent in the ‘Medium’ set that require non-trivial reasoning for human players. To judge this, we have updated Figure 2 in the paper to include levels that can be attempted by the readers/reviewers. The first is chosen specifically to be particularly simple (unfiltered test set), to give a flavour of the problems. The second is taken from the medium test set, and the third from the hard set. All three are solved by our model, we believe it illustrates the difficulty of the domain.  We encourage the reviewers to look briefly at the examples to satisfy themselves that they represent difficult planning problems.
>
> About other deep RL papers which tackle complex environments: We are certainly not the first to tackle complex environments with deep RL. But we want to emphasize 1) the combinatorial and procedural nature of our domains, which cannot be obviously solved by memorizing solutions and 2) the high level of performance achieved by the best agents (On test set distribution). To the best of our knowledge, planning abilities have not been previously demonstrated at this level by approaches that learn from environment interactions (with no access to a true model), including deep RL.
>
> Thank you for bringing to our attention the (Schmidhuber 1990) paper. We have cited this important paper in our introduction and added it as a reference for introducing the idea of multiple ticks per time-step.  We have also clarified the Werbos citation in the text.

---

### Official Review · AnonReviewer1 · 2018-11-06
**An interesting paper with strong empirical study, but unreasonable claim**

**Rating:** 5
**Confidence:** 3

**Review:**


In this paper, the authors provide an interesting view of 'planning' from the behaviorist approach. Specifically, they raise three properties as the criteria to define the planning algorithm. They exploit the convLSTM as the building block cell to construct the neural network for planning. The neural network for planning is learned with some actor critic algorithms in reinforcement learning setting. The learned policy is empirically verified to be an implicit planning module under the proposed criteria.

The paper is interesting that raise the behaviorist view of planning. The empirical study is solid showing that even not induced by some planning algorithms, the proposed neural network can still achieve better performance comparing to the neural networks which derived from special planning algorithm, e.g., VIN [1].

I pretty much like the idea that exploiting other alternative neural network structure for planning besides designing network by unfolding the existing planning algorithm.

My major concern is that their empirical study cannot support their claim that ``the planning may occur implicitly, even when the function approximator has no special inductive bias toward planning''. However, based on the experiment, this claim is not supported. In fact, the empirical results demonstrate the advantage of the proposed special structure of the neural network, i.e., DRC, rather than random existing arbitrary neural network, e.g., ResNet or LSTM. This shows that the proposed new architecture, DRC, induces special bias, although we do not explicitly know the bias yet, which is preferable to the planning tasks, even better than the VIN. In other words, the paper will be reasonable in claiming that the bias induced by VIN may be not the best and the authors provide a better alternative, rather than claiming the arbitrary flexible parametrization is enough.

Secondly, the experiments on section 4.2 for improving the performance with more computational resource is confusing. To justify the proposed algorithm can get better performance with more iteration, it should be using the learned cell in a small network, e.g., DRC(3,3), and replicate the building block to larger network, e.g., DRC(9, 9), and test the network without refinement. I am not clear what is the 'no-op' actions and how this is introduced for verifying the second criterion.

I would like to raise my score if the authors can address my questions.

[1] Aviv Tamar, Yi Wu, Garrett Thomas, Sergey Levine, and Pieter Abbeel. Value iteration networks. In Advances in Neural Information Processing Systems, pp. 2154–2162, 2016.

---

> ### Author Response · Authors · 2018-11-16
> **Response to AnonReviewer1**
>
> We’d like to thank the reviewer for their thoughtful review. We have written a common response to all three reviewers in a separate comment addressing the common issues. Below we address specific issues raised by AnonReviewer1.
>
> Our original statement about our architecture’s (lack of) inductive biases was poorly worded. What we meant to convey was: (i) we did not take inspiration from any planning algorithm and (ii) we parameterized our network as flexibly (to implement arbitrary computation) as possible. Our most powerful networks have only two relatively weak inductive biases: (a) preservation of spatial information in the hidden state of our memory modules (thus favoring ConvLSTM/ConvRNNs) and (b) repeating our network stack to allow the agent to perform extra computations on the same input with a fixed number of parameters. Both of these are obtained using off-the-shelf network components and neither is inspired by planning algorithms.
>
> Relative to other baseline architectures (VIN or ATreeC), we have fewer architectural biases, but relative to a simple ResNet or LSTM agent, our architecture admittedly has more special biases. This makes our architecture interesting in twofold ways: (a) as the reviewer agrees, the biases “induced by VIN [and ATreeC] may not be the best, and [we] provide a better alternative.” We provide empirical evidence for this on and beyond the domains these architectures were designed for. (b) Even though ResNets should be more capable of implementing generic/iterative computation, we found our architecture was more capable of planning. This certainly motivates future theoretical research on the nature of the bias our architecture induces.
>
> Thank you for noticing the lack of explanation for the no-op action (the updated version addresses this issue). There are 5 possible actions in the Sokoban environment (up, down, right, left, no-op). A no-op (no operation action) results in the agent not moving in any direction and remaining in the same environment state. For the analyses in section 4.2, we force this no-op action at the beginning of the episode to simulate an increase in the number of repeats in DRC architecture (post training). In other words, a forced no-op followed by an action on DRC(3,3) is computationally similar to a single action chosen from DRC(3, 6). By increasing the number of repeats in the DRC architecture, we leave the number of parameters unchanged and only increase the computation. Whereas, by increasing the depth, the number of parameters will be increased in addition to computation. Hence it would be infeasible to test a trained DRC(3, 3) with a DRC(9, 9) as the latter would have 6 extra untrained layers.

---

### Official Review · AnonReviewer3 · 2018-11-08
**Valuable results that don't quite support the conclusions**

**Rating:** 5
**Confidence:** 4

**Review:**

The authors describe their application of a ConvLSTM network architecture to a number of 2-D environments that have been used in the RL community to evaluate agents' capabilities for planning.

Extensive experiments show an increase in performance and generalisation compared to a number of other network architectures, including some from recent works which are designed to include an inductive bias towards implicit planning. Their model can also benefit from a deliberation phase that uses extra computation at the beginning of an episode.

The authors take this as evidence that comparatively unstructured architectures learn effective planning algorithms.

Overall, I find the paper to make a useful empirical contribution, presenting a performant architecture and results that can help guide how the community thinks about this type of benchmark. However, I believe that the framing of the results and discussion of the nature of planning should be more careful. Beyond a more careful discussion, the claims could be supported by explicit comparison to a "true" planning algorithm that makes use of a learned model.

In more depth:
(A) The empirical contributions.
The proposed architecture is described fairly clearly, and less critical elements are appropriately identified.
Comparisons to a number of planning-inspired architectures are quite comprehensive.
Experiments testing generalisation add to the body of evidence that overparameterised deep neural models can generalise well even with limited data.
I worry that the architecture may be overfit to the highly structured 2-D environments used. However, these environments are valuable testbeds whose structure is also exploited by some of the planning-inspired approaches.

(B) Conclusions & nature of planning
The authors take a behaviourist approach, identifying three properties of an "effective planning algorithm". I am not totally convinced that these are comprehensive, nor that the experiments demonstrate their clear fulfillment. This is difficult to assess because the criteria are extremely subjective.

(1) generalisation to "radically different situations". Sokoban clearly has a large state space, but it is unclear that held-out levels, or those with 7 rather than 4 boxes, are "radically" different to the training tasks.
(2) Learning from "small amounts of data". What this means is clearly subjective and highly problem-dependent.
(3) Making use of additional computation at runtime. The use of an additional deliberation phase at the beginning of the episode shows some limited scalability with computation. However, it does not permit later on-line planning to benefit from additional computation, which is a core feature of most standard planning algorithms.

Critically, the current experiments show that the large version of the proposed architecture performs better on these 3 metrics than some other architectures, but do *not* compare to anything we could unanimously agree performs "true planning".
To support the paper's claims, and to reduce the subjectivity of these metrics, it would be extremely useful to see comparisons to a "true" planning algorithm using an explicit environment model.
How many unique levels and environment interactions are required to learn a model of Sokoban? How well does that model generalise to new levels or numbers of boxes? What is the performance of e.g. MCTS using this model using different amounts of computation?
These questions could be considered out-of-scope for this particular submission, and certainly require important decisions to formulate appropriate comparisons to the model-free approach. But without at least some attempt at their answers it is hard to assess how well this model-free approach matches the behaviour of "true planning". The authors' implementation of I2A and an unspecified "powerful tree search algorithm" make me optimistic that these model-based experiments may even be feasible!
As it stands, I believe some of the claims are insufficiently supported and the overall presentation of the results overreaches.

I believe the authors could address many of these concerns and that the core contributions of the paper are valuable.
As an addendum, the Discussion section is clearer and more well supported than the framing in the Introduction.

---

> ### Author Response · Authors · 2018-11-16
> **Response to AnonReviewer3**
>
> We would like to thank the reviewer for constructive feedback. We have written a common response to all three reviewers in a separate comment addressing the common issues. Below we address specific issues raised by AnonReviewer3.
>
> About comparison to “true planning”:
>
> Let us elaborate on model-based comparisons. We provide two model-based comparisons: I2A [2] (a model-free/model-based hybrid), and a tree search algorithm, namely Levin Tree search [1]. Note that I2A also investigates MCTS (with value functions) on the same dataset. Here are our findings, which we clarify in the text. As implemented, MCTS is outperformed by Levin Tree Search and our algorithm (solving only 95% of ‘unfiltered’ levels with 2000 simulations). Now, let us compare Levin Tree Search to our algorithm. In terms of finding solutions, Levin Tree Search benefits from a systematic search property: given enough time, the solution will always eventually be found (in comparison, our model is able to leverage additional computational time - see sec. 4.2 - but is not guaranteed to find a solution). But since the amount of time required could be prohibitive (and indeed, a naive enumeration strategy has the same property), we compare performance in terms of time spent to find solutions. Here, our algorithm compares favorably: our algorithm solves 80% of hard levels in less than 48 minutes, while LTS solves 95% (using a convLSTM, similar to DRC(1,1), as policy prior) in about 10 hours. While these numbers are not directly comparable, they nevertheless strongly hint at a significantly increased computational efficiency for our approach.
>
>
> About subjectivity of evaluation:
>
> We present our three indicators as suggestive of planning from a behavioral standpoint. They are not meant to be precise criteria but rather a proposed definition to contextualize and interpret the many empirical results, as well as encourage discussion on this topic.
>
> Our experiments focus on 2D environments because we wanted to evaluate planning in the face of combinatorial complexity. A more complex state or action space could introduce confounding requirements for success (visual perception, motor control, etc) and not necessarily make the evaluation more convincing. While our own architecture achieves the state-of-the-art on a diverse set of 2D tasks (ranging from GridWorld to Sokoban, and including several planning-focussed Atari domains), and also performs favorably on our behavioral indicators of planning, it only serves to illustrate the surprising power of a simple approach. We believe our evaluation is the more central contribution of the paper in terms of enabling future research in model-based as well as model-free planning.
>
> To elaborate further on our behavioral evaluation criteria, we think that diversity and generalization abilities are well-tested with our procedurally generated levels. To solve novel Sokoban levels, even with the same number of boxes, you need to have understood something about the mechanisms of the game or encode a general reasoning strategy beyond remembering particular simple responses (e.g. “always push nearest box towards nearest goal”).  We encourage the reviewer to examine the problems pictured in our updated Figure 2 to see the difficulty in the planning required.
>
> We agree that ‘small amounts of data’ is subjective, however we compare how different approaches perform with various amounts of data. And we think it is surprising that a model-free RL agent can generalize reasonably well from training on 1k levels in Sokoban, especially in light of the community’s widespread assumption that vast amounts of training data are a prerequisite for deep RL algorithms.
>
> Lastly, on the question of making use of “additional computation at runtime,”. We emphasize this is tested after training with fixed weights. Additional computation mid-episode does not help as much because of the irreversible nature of Sokoban: if the agent has already committed to an irreversible action early in the episode, extra reasoning won’t help later. As pointed out in the results, not all recurrent architectures achieve this scalability with extra steps. While the idea of RNNs learning planning computation across time-steps is not novel, we don’t think it had been clearly demonstrated in the past. Even better scalability can surely been achieved, this was more a proof of existence here which we think is valuable.
>
>
> [1] Single-Agent Policy Tree Search With Guarantees, Orseau et al. To appear NIPS 2018
> [2] Imagination-Augmented Agents for Deep Reinforcement Learning, Weber et al. NIPS 2017

---

### Author Response · Authors · 2018-11-16
**Common response to all reviewers**

All reviewers identified our empirical results as interesting and thorough (e.g., “The empirical study is solid” AnonReviewer1, “make a useful empirical contribution” AnonReviewer3). At the same time, the reviewers disagreed with the phrasing of our claims, highlighting that some parts were too strongly worded or poorly argued. We acknowledge this and have significantly revised our introduction and abstract to be more careful in our claims about planning, about how we discuss inductive biases, and to take the reviewers’ comments into consideration.

We highlight that the word ‘planning’ is not well defined.  We lay out what we take to be behavioural characteristics of planning, but are now careful to point out that we do not have a universally agreed formal definition of planning. Thus, we cast our results more conservatively, as helping to open up discussion about findings that we found surprising (i.e. the extent to which model-free training of networks without inductive biases inspired by planning algorithms exhibit planning-like behaviours), and as important for understanding future work on planning domains. This is especially relevant given the many recent neural-planning papers.

Our empirical results show that model-free learning can produce policies that significantly outperform recent approaches that build model-based inductive biases into their architectures on challenging domains designed to test planning capacity. Our work also highlights intriguing results with deep networks trained via model-free RL. Namely, these networks come to embody policies that are surprisingly adaptive to novel scenarios, can perform better with more compute time, and can learn from limited amounts of data -- three behavioural characteristics that are more commonly associated with model-based planning. By evaluating various approaches and highlighting that an approach which uses certain classical architectures (rather than a planning-inspired architecture) can do remarkably well, our work helps open questions around the line between model-based and model-free approaches to control.

The same essential problem was identified by all reviewers (namely, being too strong in our claims in the writing). We have now addressed all of these issues with changes to the text. Given that we have done so, and that all reviewers identified our empirical contributions as strong, we’d request the reviewers to reevaluate the paper. If the reviewers have any further disagreements with our claims, please bring these to our attention quickly.

---

### Meta-Review · Area_Chair1 · 2018-12-15
**Experimental results too limited to support all claims**

**Confidence:** 4
**Recommendation:** Reject

**Metareview:**

The paper studies a convolutional LSTM (ConvLSTM) based model (DRC: Deep Repeated ConvLSTM) trained through reinforcement, and shows that it performs better than other model-free approaches, in particular in term of generalization. The ability to generalize is attributed to being able to plan. This last part is not completely convincing.

The paper is clearly written, the experiments are in 4 limited domains: Sokoban, Boxworld, MiniPacman, Gridworld. While diverse, tasks are still all similarly navigation in top-down (2D) grid worlds. It is unclear what are the limits of the reach of this study. The experimental evidence presented here could also be interpreted as: local best-response recognition of shapes (Conv) and memory of such patterns and associated actions (LSTM) are sufficient for all those environments.

Overall, this is an interesting direction, but it falls slightly short of being acceptable for publication at ICLR.